# SPECTRUM: EMPOWERING ONLINE HANDWRITING VERIFICATION VIA TEMPORAL-FREQUENCY MULTI-MODAL REPRESENTATION LEARNING

## ABSTRACT

Tapping into the uncharted multimodal representation learning in online handwriting verification (OHV), we propose SPECTRUM, a temporal-frequency synergistic model tailored to enhance handwriting representations. SPECTRUM comprises three core components: (1) a multi-scale interactor that interweaves fine-grained temporal and frequency features across multiple scales through complementary domain interaction; (2) a self-gated fusion module, dynamically integrating global temporal and frequency features via self-driven balancing. Collectively, these two components achieve micro-to-macro multimodal integration; (3) a multimodal distance-based verifier that fully harnesses temporal and frequency representations, sharpening genuine-forged discrimination beyond conventional temporal-only approaches. Extensive experiments demonstrate SPECTRUM's pronounced outperformance over existing OHV methods. Furthermore, we reveal that incorporating multiple handwritten biometrics fundamentally improves the discriminatory power of individual writing features. These findings not only validate the efficacy of multimodal learning in OHV but also encourage broader multimodal research across both feature and biometric domains, potentially opening new avenues for future explorations. Code will be publicly available.

## 1 INTRODUCTION

Evolving from quill and ink to the digital age, handwriting verification has long been a fundamental technique for identity authentication, playing crucial roles in diverse applications such as banking and legal proceedings. Generally, handwriting verification can be categorized into online and offline methods Diaz et al. (2019). Online verification utilizes dynamic data produced in the writing process such as speed and pressure for verification. In contrast, the offline counterpart analyzes digitized handwritten images obtained by scanning or photographing. This paper focuses on online handwriting verification (OHV). While signatures Tolosana et al. (2021); Lai et al. (2022) have traditionally dominated this field, recent research has expanded to include more handwritten biometrics such as isolated digits Tolosana et al. (2020a;b) or consecutive digit strings Zhang et al. (2022), broadening the realm of OHV with enhanced utility and versatility.

The development of a robust OHV system hinges on extracting powerful feature representations to capture the unique writing patterns of diverse individuals. This pursuit has driven considerable effort in enhancing the representation learning skills, such as improvements on local feature extraction Kamel et al. (2008); Jiang et al. (2022) or incorporation of global spatial attention Lai et al. (2022). Concurrently, the burgeoning field of multimodal learning has demonstrated impressive results across various domains, such as image-text alignment Radford et al. (2021); Liu et al. (2023), vision-frequency interaction Qian et al. (2020); Rao et al. (2023), and omni-modality coverage Girdhar et al. (2023); Han et al. (2024). The remarkable success of multimodal learning raises a natural question: *Could OHV similarly benefit from multimodal representation learning? If so, How?*

However, the OHV community embraces predominantly single-modal learning paradigms, particularly focusing on temporal representation learning Lai & Jin (2019); Tolosana et al. (2021); Lai et al. (2022); Jiang et al. (2022). In signal processing, frequency is an intrinsically connected modality to the time domain, often derived from the temporal sequence through techniques like Fourier transform.

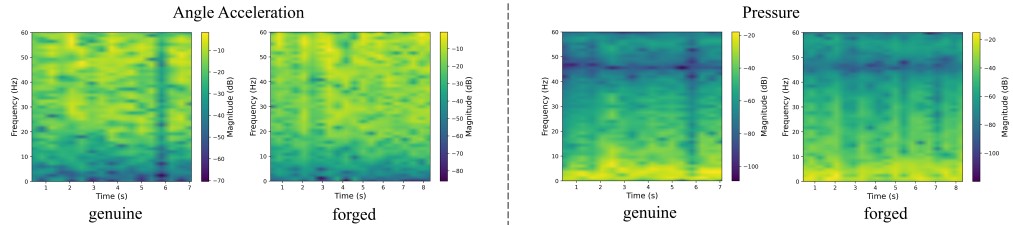

Figure 1: Spectrograms of time-domain features extracted by short-time Fourier transform (STFT) on genuine and forged handwriting samples, in which angular acceleration and pressure are taken as example features. The frequency responses showcase obvious discrepancies between genuine and forged handwriting, which could enrich the temporal features for multimodal discrimination.

The potency of frequency modeling has been verified in other forgery verification fields, such as face forgery detection Qian et al. (2020); Miao et al. (2023) and speaker verification Liu et al. (2022; 2024). Similarly, in online handwriting analysis, frequency features provide a unique discriminatory perspective, capturing crucial writing traits like rhythms and periodicities that aid in distinguishing genuine from forged samples. As shown in Fig. 1, the spectrograms extracted by short-time Fourier transform (STFT) reveal significant discrepancies between genuine and forged samples in the frequency domain. Despite the potential of frequency modeling, its exploration in OHV has been confined to being used as a sole, superficial feature extraction stage without further modeling Nakanishi et al. (2006); Nanni & Lumini (2008), resulting in limited feature representation capabilities. In addition, prior studies suffer from modality isolation, myopically focusing on either the temporal or frequency domains but overlooking the potential integration of both. A multimodal approach that synergizes temporal and frequency features would unlock more discriminative handwriting representations, intuitively suggesting a compelling direction for developing better OHV systems.

Motivated by this insight, we propose **SPECTRUM**, a **SPEC**tral-**T**empo**R**al **U**nified **M**odel that integrates temporal and frequency modalities for multimodal online handwriting verification. First, we devise two components to achieve micro-to-macro multimodal integration ($\mathbf{M}^4\mathbf{I}$) across temporal and frequency domains. (1) *Micro integration.* We propose a multi-scale interactor to facilitate fine-grained interaction between temporal and frequency features. Handwriting sequences are split into even and odd sub-sequences for independent temporal and frequency analyses. We utilize a projection layer to preserve the temporal features, while formulating the frequency modeling by combining a 1D (inverse) Fourier transform with learnable complex weights of scale $l$ to emphasize salient frequency features Rao et al. (2023). The two sub-sequences are then recombined to enable mixed-domain interaction. By varying scale $l$, we develop the multi-scale interactor to aggregate multi-scale contexts for scale-reciprocal complementations. (2) *Macro integration.* We introduce a self-gated fusion module that dynamically weights the contributions of global temporal and frequency features, attempting for self-optimized feature fusion. Collectively, these two modules accomplish temporal and frequency integration in a micro-to-macro manner, ensuring comprehensive multimodal interplay. Second, we propose a multimodal distance-based verifier (**MDV**), which combines Dynamic Time Warping distance computed with temporal features and Euclidean distance with frequency features to enhance discrimination between genuine and forged samples. It naturally harnesses representations of both modalities under a unified multimodal framework, transcending the reliance on merely temporal features in prior works and resulting in better verification accuracy.

We evaluate the proposed SPECTRUM using three online handwriting datasets: MSDS-ChS Zhang et al. (2022) (Chinese Signature), MSDS-TDS Zhang et al. (2022) (Token Digit String (TDS)), and DeepSignDB Tolosana et al. (2021) (Latin Signature). Experiments demonstrate a pronounced outperformance of our model over state-of-the-art OHV methods that solely depend on temporal representation learning, evidencing the effectiveness of the $\mathrm{M}^4\mathrm{I}$ mechanism and MDV in incorporating frequency features for multimodal learning. In addition, we investigate multimodal fusion between multiple biometric modalities, where the Chinese signature and TDS are combined to enrich individual writing representations. This approach yields further improvements in verification performance, suggesting that multimodal learning can be extended across not only feature domains (temporal and frequency) but also biometric domains (Chinese signature and TDS), potentially opening new avenues for further research in this field. Our main contributions are summarized as follows:

- We propose SPECTRUM, a multimodal representation model for OHV, empowering traditional temporal modeling with frequency analysis to enable effective multimodal learning.
- SPECTRUM features a multi-scale interactor and self-gated fusion module, infusing the model with micro-to-macro multimodal feature integration. In addition, we design a multimodal distance-based verifier to enhance verification by naturally leveraging both temporal and frequency representations within the multimodal context.
- Experiments demonstrate the superiority of SPECTRUM over existing OHV methods. We further reveal the effectiveness of incorporating multiple biometric modalities to enhance representation discrimination and OHV performance, potentially inspiring future research.

## 2 RELATED WORK

**Online Handwriting Verification Techniques**. Online handwriting verification (OHV) has seen substantial progress in recent decades, primarily focusing on online signature verification Diaz et al. (2019) due to its pervasive usage. This technique typically constitutes two stages: feature representation and decision making. (1) *Feature representation.* The evolution from traditional hand-crafted extraction methods Sharma & Sundaram (2017); Tang et al. (2018); Farimani & Jahan (2018); Okawa (2021) to modern deep learning methods has established new state-of-the-art performance. Current deep learning approaches broadly operate in two paradigms. The first type concentrates on local feature modeling, often developed in conjunction with Dynamic Time Warping (DTW). PSN Wu et al. (2019b) and TA-RNNs Tolosana et al. (2021) pre-align handwriting sequences using DTW before inputting them to CNN/RNN-based models. DeepDTW Wu et al. (2019a) uses a DTW on top of a Siamese CNN to enhance local invariance learning. RAN Lai & Jin (2019) proposes a length-normalized path signature descriptor to describe local signature trajectories. DsDTW Jiang et al. (2022) integrates the differentiable soft-DTW into the loss function to improve local discriminative learning. The second paradigm captures global representations. Park et al. Park et al. (2019) utilize an LSTM-CNN network to analyze features at both stroke and signature levels. Li et al. Li et al. (2019) progressively model the stroke features and the holistic signature with RNN. Sig2Vec Lai et al. (2022) proposes a selective pooling module to pool the temporal sequence, converging subspace features into a fixed-length vector with global context awareness. (2) *Decision making.* Typically configured in an open-set manner, OHV systems are trained on limited data but tested on unlimited unseen data. This requires models to generate feature vectors to assess similarities between templates and queries, thereby verifying queries' authenticity. Common approaches include Euclidean/DTW distance-based verifiers that authenticate queries falling within specific thresholds Lai & Jin (2019); Lai et al. (2022); Jiang et al. (2022), subject-independent classifiers evaluating sample-wise distances Wu et al. (2019a;b), and sigmoid scoring based on pre-given thresholds Tolosana et al. (2021).

Recently, the OHV field has expanded beyond signatures to encompass emerging handwritten biometrics like digit/digit strings. Tolosana et al. propose the e-BioDigit Tolosana et al. (2020a) and MobileTouchDB Tolosana et al. (2020b) datasets for second-level identity authentication using separate digits. Zhang et al. Zhang et al. (2022) propose the MSDS dataset, including the MSDS-ChS and MSDS-TDS subsets. This work demonstrates that mainstream signature verification methods can be seamlessly transferred to other handwritten biometrics, such as Token Digit String (TDS).

**Frequency Learning for Online Handwriting Verification.** While contemporary methods primarily rely on the temporal domain for handwriting analysis, earlier research has explored frequency analysis for handwriting characterization due to the straightforward transformation from temporal to frequency domain. The Wavelet transform Nakanishi et al. (2006); Nanni & Lumini (2008); Fahmy (2010); Chavan et al. (2017); Yang & Liu (2017); Miaba et al. (2018); Alpar (2018) and Fourier transform hua Quan et al. (2006); Yanikoglu & Kholmatov (2009); Chavan et al. (2017); Miaba et al. (2018) are mostly adopted, while additional frequency features such as Discrete Cosine/Hartley/Walsh-Hadamard/Kreke/Mellin transform Nanni & Lumini (2008); Chavan et al. (2017); Fallah et al. (2011) are also explored. Despite these efforts, frequency learning for OHV has been shackled by two critical drawbacks. (1) *Limited feature extraction.* Most studies rely solely on frequency transforms for feature extraction without further modeling, usually yielding insufficiently discriminative features. (2) *Modality isolation.* Prior methods rely exclusively on the frequency modality but overlook the potential synergy with temporal modeling, which is an oversight that also persists in current cutting-edge temporal-centric approaches. To address these issues, we propose SPECTRUM, a multimodal

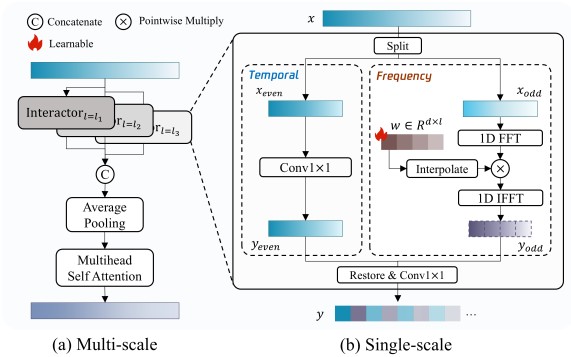

Figure 2: Overall framework of the proposed SPECTRUM. **Top**: Model training process. **Middle**: Detailed architecture of SPECTRUM, which performs micro-to-macro multimodal learning through two stacked M⁴I blocks. The multi-scale interactor in the last M⁴I block exclusively outputs frequency features, which are pooled to yield $f_F$. **Bottom**: Model inference (verification) process, where MDV harnesses both temporal and frequency representations to enhance verification robustness.

learning model that interweaves temporal and frequency in a micro-to-macro integration paradigm, empowering handwriting representation from the multimodal perspective.

## 3 METHODOLOGY

Fig. 2 illustrates the overall framework of the proposed SPECTRUM. Our model synergizes temporal and frequency domains through the multi-scale interactor and self-gated fusion module (Sec. 3.1), while using the multimodal distance-based verifier (MDV) (Sec. 3.2) to enhance verification.

### 3.1 MICRO-TO-MARCO MULTIMODAL INTEGRATION (M⁴I) MECHANISM

To fully combine temporal and frequency features, we propose the multi-scale interactor and the self-gated fusion module for micro-to-macro multimodal integration (M⁴I), which corresponds to the M⁴I blocks depicted in Fig. 2.

**Micro-level multimodal learning**. We design a multi-scale interactor to capture fine-grained interactions between temporal and frequency features. The multi-scale interactor fundamentally consists of multiple single-scale interactors, with architectures detailed in Fig. 3 (a) and Fig. 3 (b) for each. We begin with deliberating on the design

Figure 3: Schematic of the multi-scale interactor.

of a single-scale interactor. Given an input temporal handwriting sequence $x \in \mathbb{R}^{d \times L}$ ($d$ is the embedding dimension and $L$ is the sequence length), we split it into two sub-sequences $x_{even} \in \mathbb{R}^{d \times \lceil L/2 \rceil}$ and $x_{odd} \in \mathbb{R}^{d \times \lceil L/2 \rceil}$ by separating even and odd timesteps along the spatial dimension. $x_{even}$ is dedicated to preserving information and undergoes a simple $1 \times 1$ convolution to derive $y_{even}$. In contrast, $x_{odd}$ is assigned for frequency modeling. Inspired by Rao et al. (2023), we perform 1D discrete Fourier transform (DFT) on $x_{odd}$ to calculate its spectrum response $X$. Given each embedding dimension $i \in [0, d-1]_{\mathbb{Z}}$, the frequency response $X[i]$ for $x_{odd}[i]$ is calculated as:

$$X[i, k] = \sum_{n=0}^{N-1} x_{odd}[i, n] e^{-j \frac{2\pi k}{N} n} \in \mathbb{R}^{1 \times N}, k \in [0, N-1]_{\mathbb{Z}}, \tag{1}$$

where $N = \lceil L/2 \rceil$, $j$ is the imaginary unit, and $X[k]$ represents the frequency response of $x[n]$ at the frequency point $\omega_k = \frac{2\pi k}{N}$. By aggregating $X[i]$, we can obtain the entire frequency features $X = \{X[i]\} \in \mathbb{R}^{d \times N}$. For real-valude inputs $x[i, n]$, its DFT response is inherently symmetric Dubois & Venetsanopoulos (1978); Rao et al. (2023), $i.e.$, $X[i, N-k] = X^*[i, k]$. Therefore, the

half of DFT, *i.e.*, $\hat{X} = \{X[i,k]\} \in \mathbb{R}^{d \times \lceil N/2 \rceil}, k \in [0, \lceil N/2 \rceil - 1]_{\mathbb{Z}}$, is sufficient to cover the full frequency characteristics of $x[n]$.

Subsequently, we introduce a 1D learnable complex weights $w \in \mathbb{R}^{d \times l}$, designed to modulate the frequency features $\hat{X}$ and selectively amplify the discriminative aspects. The length $l$ of $w$ reflects the "scale" term of the single-scale interactor. However, the predefined length $l$ in the model configuration may not match the spectral length $N$. To reconcile this, we first interpolate $w$ to length $\lceil N/2 \rceil$ using bilinear interpolation, resulting in $\bar{w}$, and then multiply it with $\hat{X}$:

$$\bar{w} = interpolate(w, \lceil N/2 \rceil),$$
$$\bar{X} = \hat{X} \odot \bar{w}, \tag{2}$$

where $\odot$ denotes point-wise multiplication. With the modulated frequency features, we perform 1D inverse Discrete Fourier transform (IDFT) on $\bar{X}[i]$ of each embedding dimension $i$. Since $\bar{X}[i]$ represents the half-spectrum due to the conjugate symmetry, we first reconstruct it to the full-spectrum $\tilde{X}[i]$ and then perform the IDFT:

$$\tilde{X}[i,k] = \begin{cases} \bar{X}[i,k], 0 \leq k < \lceil N/2 \rceil, \\ \bar{X}^*[i, N-k], \lceil N/2 \rceil \leq k < N, \end{cases}$$
$$y_{odd}[i,n] = \frac{1}{N} \sum_{k=0}^{N-1} \tilde{X}[i,k] e^{j\frac{2\pi k}{N}n} \in \mathbb{R}^{d \times N}. \tag{3}$$

Here, we derive the remapped output $y_{odd}$, representing frequency-modulated writing features. In implementation, we adopt the more efficient while functionally equivalent Fast Fourier transform (FFT) and inverse Fast Fourier transform (IFFT) to compute DFT and IDFT. This reduces the computation complexity from $\mathcal{O}(N^2)$ to $\mathcal{O}(N log N)$, essentially expediting both training and inference.

Given the temporal output $y_{even}$ and frequency output $y_{odd}$, we restore them to a new sequence according to their original even and odd positions to interwind the temporal and frequency features. The interleaved features are then passed through a $1 \times 1$ convolution to derive the output $y$ of a single-scale interactor. Afterward, we build the multi-scale interactor by using $m$ single-scale interactors with varying scales $l$, feeding the input $x$ to each of them and consolidating their output by average pooling. We further impose a standard multi-head self-attention Vaswani et al. (2017) to the averaged sequence and obtain the final mixed-modality output. $m$ is empirically set to 3.

**Macro-level multimodal learning.** As shown in Fig. 2, the temporal features passed through the convolution module (Conv) are fed into the multi-scale interactor for fine-grained temporal-frequency learning. More globally, the frequency-modulated features can be further fused with the external temporal features. To this end, we introduce a self-gated fusion module for global multimodal interaction as illustrated in Fig. 4. Given temporal features $f_{time} \in \mathbb{R}^{L \times d}$ and frequency features $f_{freq} \in \mathbb{R}^{L \times d}$, they are concatenated along the channel dimension to yield $f \in \mathbb{R}^{L \times 2d}$. We then compute a gate coefficient $g$ to dynamically fuse the two modality features:

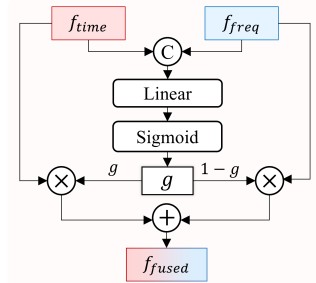

$$g = f@W^T + b, W \in \mathbb{R}^{d \times 2d}, b \in \mathbb{R}^d,$$
$$f_{fused} = f_{time} \odot g \oplus f_{freq} \odot (1-g), \tag{4}$$

Figure 4: Schematic of the self-gated fusion module.

where @ denotes matrix multiplication, $\odot$ signifies point-wise multiplication, $\oplus$ signifies point-wise addition, $W$ and $b$ are weights and biases of a linear layer. The fused features $f_{fused}$ are combined by adaptively weighting the contributions of temporal and frequency features through the self-derived gate $g$, accomplishing global multimodal feature integration.

**Discussion.** In the multi-scale interactor, the segmented sub-sequences $x_{even}$ and $x_{odd}$ retain much of the original sequence's dynamic and structural integrity despite the reduced resolution, ensuring sufficient fundamental handwriting characteristics for subsequent temporal and frequency analyses. Our frequency modeling approach closely follows Rao et al. (2023), but is tailored specifically for 1D handwriting sequences rather than 2D images. Through $x_{even}$'s transformation into the frequency domain and the modulation of learnable weights, our model adaptively emphasizes the unique writing

patterns among specific frequency bands while filtering out noise. The following recombination naturally interweaves the temporal and frequency sequences, promoting deep interaction and complementarity between the two modalities. Furthermore, the self-gated fusion facilitates a more holistic multimodal consolidation with self-driven feature balance. These designs collaboratively enable a comprehensive micro-to-macro integration of temporal and frequency features.

## 3.2 Multimodal Distance-Based Verifier

Similar to Sig2Vec Lai et al. (2022) and DsDTW Jiang et al. (2022), SPECTRUM exploits a distance-based verifier that compares the feature representations of template and query handwriting for verification. Nevertheless, prior methods are confined to solely utilizing temporal embeddings. Given the dual temporal and frequency awareness in our SPECTRUM, we propose a multimodal distance-based verifier (MDV) to leverage representations from both modalities for enhanced discrimination. As shown in the right panel of Fig. 2, given two handwriting $x^i$ and $x^j$, they undergo model feature extraction $\phi$ and derive the temporal feature sequences $f_T^i, f_T^j \in \mathbb{R}^{L_T \times d}$ and frequency feature vectors $f_F^i, f_F^j \in \mathbb{R}^d$ ($L_T$ is the sequence length and $d$ is the embedding dimension). We compute the Dynamic Time Warping (DTW) distance between temporal sequences and Euclidean distance between frequency vectors as:

$$
\begin{aligned}
d_T(x^i, x^j) &= DTW(\phi(x^i), \phi(x^j)) = DTW(f_T^i, f_T^j), \\
d_F(x^i, x^j) &= ||\phi(x^i) - \phi(x^j)||^2 = ||f_F^i - f_F^j||^2,
\end{aligned}
\tag{5}
$$

Given $n$ template handwriting $\{x_u^1, ..., x_u^n\}$ attributed to writer $u$, we compute average pairwise distance between their temporal features, denoted as $\bar{d_T^u}$ ($\bar{d_T^u} = 1$ if $n = 1$). For a query handwriting $x^q$ claiming to be writer $u$, we compute temporal and frequency scores between $x^q$ and all templates:

$$
\begin{aligned}
s_T^{p,u}(x^q) &= d_T(x_u^p, x^q)/\sqrt{\bar{d_T^u}}, \\
s_F^{p,u}(x^q) &= d_F(x_u^p, x^q)/\sqrt{\bar{d_T^u}},
\end{aligned}
\tag{6}
$$

where $p \in [1, n]_{\mathbb{Z}}$. After acquiring all scores, we can compute the mean and minima of the temporal scores $s_T^{u_{avg}}$, $s_T^{u_{min}}$, and frequency scores $s_F^{u_{avg}}$, $s_F^{u_{min}}$. Then, we use the frequency scores to adaptively weight the temporal scores, determining whether to accept the query by:

$$
s_T^{u_{min}}(1 + sigmoid(s_F^{u_{min}})) + s_T^{u_{avg}}(1 - sigmoid(s_F^{u_{avg}})) < c,
\tag{7}
$$

where $c$ is a pre-set threshold. If the distance summation fulfills Eq. 7, the query $x^q$ is deemed genuine for writer $u$ and accepted, otherwise it is determined as a forgery and rejected. By varying the threshold $c$, we can compute the Equal Error Rate metric (Sec. 4.1) for performance evaluation.

By harmonizing both temporal and frequency representations, MDV naturally fits in the multimodal framework of SPECTRUM and sharpens the distinction between genuine samples and forgeries. This dual action transcends the limitation of solely using temporal features for verification in previous studies. Eq. 7 implies enhancing the more discriminative temporal scores while minimizing the less influential ones by adaptively re-weighting temporal scores with frequency scores. Importantly, the weights are dynamically derived from frequency features rather than manually set, ensuring flexible and robust adaptation to diverse handwriting scenarios.

## 3.3 Model Optimization

As described in Sec. 2 and Sec. 3.2, SPECTRUM performs open-set verification using the output temporal and frequency feature representations. Therefore, we adopt metric-learning to optimize the model's representation ability. As shown in Fig. 2, $f_T \in \mathbb{R}^{L_T \times 64}$ denotes the temporal features of an input processed by GRU and Head following two M$^4$I blocks. The Head module is a multi-layer perceptron. We then input $f_T$ into a lifted-structure triplet loss Oh Song et al. (2016) to separate genuine samples and forged samples in the embedding-space, where we use soft-DTW ($\gamma = 5$) as the inner distance function following DsDTW Jiang et al. (2022). The computational details are also similar to DsDTW, which yields an intra-writer variation term $\mathcal{L}_{intra}$ controlled by a parameter $\lambda$ and a vanilla triplet loss term $\mathcal{L}_{tri}$. In addition, the frequency features $f_F$ of the last multi-scale interactor are compressed by a selective pooling layer Lai et al. (2022) and the Head module into binary logits.

The logits are supervised by a binary cross entropy loss $\mathcal{L}_{BCE}$ (genuine sample→label 1; forged sample→label 0). The full optimization objective is formulated as:

$$\mathcal{L} = \lambda\mathcal{L}_{intra} + \mathcal{L}_{tri} + \mathcal{L}_{BCE}. \tag{8}$$

# 4 EXPERIMENT

## 4.1 EXPERIMENT PROTOCOL

**Dataset.** We assess SPECTRUM on three OHV datasets, including the two subsets of the MSDS dataset Zhang et al. (2022), *i.e.*, MSDS-ChS (Chinese signature), MSDS-TDS (Token Digit String), and DeepSignDB Tolosana et al. (2021) (Latin signature). These datasets are the currently largest public datasets for their respective handwriting type. We split the data to ensure that testing user data is entirely unseen during training, adhering to the open-set setting. MSDS-ChS and MSDS-TDS share the same 402 writers. As per Zhang et al. (2022), we divide the first 202 individuals as the training set while assigning the rest 200 users as the testing set. This results in 8,080/8000 training/testing samples from 202/200 users for each dataset. We use the two-session (across-session) data of each dataset by default. DeepSignDB consists of five subsets, in which, however, the Biosecure DS2 subset Ortega-Garcia et al. (2010) currently releases only training data but not testing data. To ensure fair comparisons, we follow Lai et al. (2022); Jiang et al. (2022) and utilize the same subsets as them during training and testing, where the official "development" and "evaluation" sets of all subsets are utilized as training/testing data, respectively. This results in 21,104/20,596 training/testing samples from 528/512 users. We perform data preprocessing on the online handwritten data and extract 15 time-function features as input to the models, as detailed in Appendix A.

**Metric.** We adopt Equal Error Rate (EER) as the evaluation metric, which refers to the point where False Acceptance Rate equals False Rejection Rate. The proposed MDV is employed to compute EER%, with details provided in Sec. 3.2. Following the original papers of MSDS and DeepSignDB, we report EERs under both a global threshold and a local (user-specific) threshold and display the results in the format of $EER_g/EER_l$ on MSDS-ChS and MSDS-TDS, while reporting only EERs under the global threshold on DeepSignDB. All results are reported in percentage.

**Impostor types.** We consider both skilled and random forgeries as impostor types. Skilled forgeries are selected from the skillfully forged samples of each user that are originally provided in the datasets, while random forgeries are selected from the genuine samples of other users.

**Template selection.** The number of genuine handwriting templates used during verification essentially affects model performance. For MSDS-ChS and MSDS-TDS, we follow the original paper and use one to four templates against one query in skilled forgery verification, denoted as 4 vs 1, 3 vs 1, 2 vs 1, and 1 vs 1; while using four and one templates in random forgery verification. For DeepSignDB, we follow the original paper to employ four and one templates in verification for both skilled forgery and random forgery scenarios. To guarantee the reproducibility of test results, we consistently take the first $n$ samples among all genuine samples of the user as templates.

## 4.2 COMPARISON WITH STATE-OF-THE-ART METHOD

We compare the handwriting verification performance of SPECTRUM against existing state-of-the-art (SOTA) methods on MSDS-ChS, MSDS-TDS, and DeepSignDB, with results respectively summarized in Tables 1 to 3. DTW Vintsyuk (1968) denotes directly computing Dynamic Time Warping distance on the input time-function features for handwriting matching without training, while other methods are all trained models. From the results, we draw the following observations.

(1) As evidenced in Tables 1 and 2, SPECTRUM surpasses existing methods in most cases on the MSDS-ChS and MSDS-TDS datasets. Under skilled forgery scenarios, it achieves EERs of 5.30/2.47 ($EER_g/EER_l$) on MSDS-ChS and 3.38/1.20 on MSDS-TDS, outperforming the second-best performance of 5.91/2.90 and 4.13/1.42 by significant margins, substantiating its superiority. Under random forgery scenarios, SPECTRUM outstrips other models like DsDTW and Sig2Vec on both datasets, especially on MSDS-TDS. Although the DTW method slightly edges out SPECTRUM, the margin is narrow, underscoring SPECTRUM's competitiveness. The outperformance primarily stems from the multimodal learning that integrates both temporal and frequency domains, imbuing SPECTRUM with more powerful handwriting representation ability than other single-modal methods.

Table 1: Comparison of SPECTRUM and existing OHV methods on MSDS-ChS Zhang et al. (2022). The best results are marked in **bold** and the second-best results are marked with underline.

| Method | Venue | Skilled Forgery ↓ | | | | Random Forgery ↓ | |
|---|---|---|---|---|---|---|---|
| | | 4 vs 1 | 3 vs 1 | 2 vs 1 | 1 vs 1 | 4 vs 1 | 1 vs 1 |
| DTW Vintsyuk (1968) | - | 11.66/7.70 | 11.37/7.44 | 12.42/7.26 | 17.26/8.93 | **0.58**/0.20 | **1.03**/0.27 |
| DeepDTW Wu et al. (2019a) | ICDAR'19 | 7.14/3.70 | 7.16/3.71 | 7.53/3.71 | 12.60/4.77 | 0.61/0.16 | 5.41/1.10 |
| TA-RNNs Tolosana et al. (2021) | TBIOM'21 | 7.69/5.22 | 7.91/5.67 | 8.34/6.36 | 9.04/5.05 | 2.67/0.47 | 1.55/0.57 |
| Sig2Vec Lai et al. (2022) | TPAMI'22 | 9.03/4.97 | 8.78/4.92 | 9.87/5.16 | 15.10/7.27 | 1.93/0.74 | 5.09/1.18 |
| DsDTW Jiang et al. (2022) | TIFS'22 | 5.91/2.90 | 5.69/2.90 | 5.96/2.77 | **9.58**/**3.99** | 0.84/**0.11** | 1.87/**0.17** |
| SPECTRUM (**Ours**) | This Work | **5.30**/**2.47** | **5.33**/**2.53** | **5.88**/**2.62** | 10.70/4.97 | 0.72/**0.11** | 2.72/0.32 |

Table 2: Comparison of SPECTRUM and existing OHV methods on MSDS-TDS Zhang et al. (2022).

| Method | Venue | Skilled Forgery ↓ | | | | Random Forgery ↓ | |
|---|---|---|---|---|---|---|---|
| | | 4 vs 1 | 3 vs 1 | 2 vs 1 | 1 vs 1 | 4 vs 1 | 1 vs 1 |
| DTW Vintsyuk (1968) | - | 9.99/5.75 | 9.94/5.78 | 10.01/5.95 | 14.46/6.76 | **0.25**/**0.01** | **0.30**/0.04 |
| DeepDTW Wu et al. (2019a) | ICDAR'19 | 5.75/1.94 | 5.60/1.93 | 5.49/1.95 | 9.56/2.11 | 0.63/0.28 | 5.16/0.40 |
| TA-RNNs Tolosana et al. (2021) | TBIOM'21 | 5.11/2.91 | 5.44/3.06 | 5.77/3.16 | 5.94/2.60 | 1.71/0.40 | 0.85/0.21 |
| Sig2Vec Lai et al. (2022) | TPAMI'22 | 5.18/2.07 | 5.24/2.22 | 5.94/2.17 | 7.01/3.26 | 1.66/0.26 | 1.76/0.28 |
| DsDTW Jiang et al. (2022) | TIFS'22 | 4.13/1.42 | 4.05/1.41 | 4.40/1.32 | 5.76/**1.85** | 0.42/0.07 | 0.59/0.14 |
| SPECTRUM (**Ours**) | This Work | **3.38**/**1.20** | **3.48**/**1.11** | **3.57**/**1.18** | **5.20**/2.10 | 0.30/0.04 | 0.76/**0.02** |

(2) Table 3 demonstrates that SPECTRUM delivers generally comparable performance compared to the SOTA methods on the DeepSignDB dataset. Although the Sig2Vec model primarily holds sway, our SPECTRUM exhibits the best/second-best results in some cases, such as in the skilled forgery verification based on stylus-/finger-written signatures. However, we can observe a notable performance decline on random forgery verification under the finger scenario. This suggests that, for finger-written Latin signatures, frequency features could benefit discerning genuine samples against skilled forgeries but may be less effective in distinguishing genuine samples of different writers.

(3) Our model exhibits better performance on the MSDS-TDS dataset than on the MSDS-ChS dataset, resonating with the phenomenon discovered in Zhang et al. (2022) that the accuracies of TDS verification are higher than those of Chinese signature verification. Importantly, the MSDS-ChS and MSDS-TDS are collected from the same 402 users and share identical user data splitting, ensuring fair comparisons. This performance consistency reinforces that TDS could be a more effective and reliable handwritten identifier than Chinese signature, prompting us to pay more attention to this biometric medium.

(4) Compared to DeepSignDB, verification performances on MSDS-ChS and MSDS reveal more room for improvement, indicating that Chinese signature and TDS could be more challenging handwritten biometrics than Latin signature. Our model significantly improves EERs on MSDS-ChS and MSDS-TDS, particularly the latter one, narrowing the performance gap between the emerging TDS and traditional Latin signature and bolstering its applicability. This not only demonstrates the effectiveness of SPECTRUM but also advances the adoption of more emerging handwritten biometrics for more practical OHV.

## 4.3 ABLATION STUDY

We conduct ablation studies on the MSDS-TDS and MSDS-ChS datasets to investigate the effectiveness of individual components in the proposed SPECTRUM. *Baseline* indicates a model consists of merely two Conv modules (Fig. 2) and a GRU. *Frequency* refers to incorporating a single-scale interactor for frequency modeling along with the basic temporal modeling. ✗ indicates the removal of specific modules, except for replacing the self-gated fusion module with an addition operation. Results are summarized in Table 4.

Comparing lines 1 and 2, we observe that the initial incorporation of a single-scale interactor for frequency modeling impairs model performance. However, lines 3-4 reveal that introducing the

Table 3: Comparison of SPECTRUM and existing OHV methods on DeepSignDB Tolosana et al. (2021).

| Method | Venue | Stylus | | | | Finger | | | |
| | | Skilled Forgery ↓ | | Random Forgery ↓ | | Skilled Forgery ↓ | | Random Forgery ↓ | |
| | | 4 vs 1 | 1 vs 1 | 4 vs 1 | 1 vs 1 | 4 vs 1 | 1 vs 1 | 4 vs 1 | 1 vs 1 |
|---|---|---|---|---|---|---|---|---|---|
| DTW Vintsyuk (1968) | - | 4.53 | 7.06 | 1.23 | 1.98 | 10.66 | 14.74 | 1.02 | 1.25 |
| TA-RNNs Tolosana et al. (2021) | TBIOM'21 | 3.30 | 4.20 | 0.60 | 1.50 | 11.30 | 13.80 | 1.00 | 1.80 |
| Sig2Vec Lai et al. (2022) | TPAMI'22 | **2.54** | 4.08 | **0.48** | **0.84** | 6.97 | **10.87** | **0.79** | **1.86** |
| DsDTW Jiang et al. (2022) | TIFS'22 | **2.54** | **4.04** | 0.97 | 1.69 | 6.99 | 11.84 | 1.81 | 2.89 |
| SPECTRUM (**Ours**) | This Work | 2.61 | 4.31 | 1.13 | 1.99 | **6.96** | 11.44 | 2.38 | 4.63 |

Table 4: Ablation study on MSDS-TDS Zhang et al. (2022) and MSDS-ChS Zhang et al. (2022). *Baseline* indicates a model consists of merely two Conv modules (Fig. 2) and a GRU. *Frequency* denotes introducing a single-scale interactor for frequency modeling. ✗ for the self-gated fusion module denotes replacing it with an addition.

| Line | Baseline | Frequency | Multi-Scale | Self-Gated Fusion | MDV | MSDS-TDS | | | | MSDS-ChS | | | |
| | | | | | | Skilled Forgery ↓ | | Random Forgery ↓ | | Skilled Forgery ↓ | | Random Forgery ↓ | |
| | | | | | | 4 vs 1 | 1 vs 1 | 4 vs 1 | 1 vs 1 | 4 vs 1 | 1 vs 1 | 4 vs 1 | 1 vs 1 |
|---|---|---|---|---|---|---|---|---|---|---|---|---|---|
| 1 | ✓ | ✗ | ✗ | ✗ | ✗ | 4.13/1.30 | 6.09/2.09 | 0.36/0.05 | 1.21/0.08 | 5.98/2.80 | 11.30/5.13 | 1.19/0.22 | 4.25/0.57 |
| 2 | ✓ | ✓ | ✗ | ✗ | ✗ | 5.02/1.38 | 7.28/2.39 | 0.49/0.08 | 1.39/0.09 | 6.50/2.91 | 11.22/4.94 | 0.98/0.14 | 3.35/0.36 |
| 3 | ✓ | ✓ | ✗ | ✗ | ✓ | 4.95/1.36 | 7.28/2.39 | 0.50/0.09 | 1.39/0.09 | 6.13/2.86 | 11.22/4.94 | 0.93/0.15 | 3.35/0.36 |
| 4 | ✓ | ✓ | ✓ | ✗ | ✓ | 4.05/1.43 | 5.90/2.07 | 0.34/0.04 | **0.70**/0.03 | 5.49/2.45 | 10.40/4.68 | 0.90/0.17 | 3.22/0.47 |
| 5 | ✓ | ✓ | ✗ | ✓ | ✓ | 4.67/1.46 | 7.02/2.25 | 0.59/0.05 | 1.54/0.08 | 6.20/3.12 | 12.33/5.85 | 1.05/0.14 | 3.96/0.54 |
| 6 | ✓ | ✓ | ✓ | ✓ | ✗ | 3.44/1.22 | 5.20/2.10 | **0.25/0.04** | 0.76/0.02 | 5.51/2.75 | 10.70/4.97 | 0.74/**0.10** | 2.72/0.32 |
| 7 | ✓ | ✓ | ✓ | ✓ | ✓ | **3.38/1.20** | **5.20/2.10** | 0.30/**0.04** | 0.76/0.02 | **5.30/2.47** | **10.70/4.97** | **0.72**/0.11 | **2.72/0.32** |

multi-scale interactor rather than the single-scale one significantly improves model performance, evidenced by the gains of 0.90%/0.64% (global threshold) in the most difficult skilled forgery scenario on MSDS-TDS and MSDS-ChS, respectively. Furthermore, comparing lines 5 and 7, removing the multi-scale interactor from the entire model results in 1.29% and 0.90% declines (global threshold; skilled forgery; the same as follows) on MSDS-TDS and MSDS-ChS. These outcomes strongly demonstrate the significance of the multi-scale interactor in introducing fine-grained frequency features and enhancing stylistic representations. In addition, the self-gated fusion module brings 0.67% and 0.19% improvements on the two datasets, respectively (lines 4 and 7). The MDV further boosts performance by 0.06% and 0.21% (lines 6 and 7) on two datasets. Notably, incorporating all our designs leads to the best overall performance. The ablation results substantiate the effectiveness of the modules in SPECTRUM, validating the enhanced representation performance brought about by our multimodal learning approach.

## 4.4 BIOMETRIC-BASED MULTIMODAL REPRESENTATION LEARNING

We further investigate multimodal learning from the perspective of multiple biometric mediums. Since the Chinese signature (ChS) in MSDS-ChS Zhang et al. (2022) and Token Digit String (TDS) Zhang et al. (2022) in MSDS-TDS come from the same writers, it offers a natural avenue to incorporate both ChS and TDS to explore their collaborative potential for OHV. Therefore, we construct a dual-path model, in which both paths leverage identical established models but respectively receive ChS and TDS as inputs. Two established OHV models and the proposed SPECTRUM are applied in this dual-path architecture for experiments. We concatenate sequence representations along spatial dimensions or average logits from the two paths for optimization and testing. The data of MSDS-ChS and MSDS-TDS is merged, following the split in Sec. 4.1, to create consolidated training and testing sets while maintaining the open-set setting. Experimental results are presented in Table 5.

As observed, on the three methods, combining ChS and TDS generally strengthens performance compared to employing either modality alone, particularly in the most challenging skilled forgery scenario. These improvements bring forth several inspirations. (1) Simultaneously utilizing multiple handwritten biometrics indeed improves verification performance. The improvement is likely due to the richer feature set obtained by combining two biometrics, which essentially amplifies the stylistic representations of individuals and enhances the discriminatory power. (2) Under the combined-biometric context, SPECTRUM attains consistently optimal results in skilled forgery verification and

Table 5: Multimodal fusion between two biometrics: Chinese signature and Token Digit String, using data from MSDS-ChS Zhang et al. (2022) and MSDS-TDS Zhang et al. (2022), respectively.

| Method | Biometric | Skilled Forgery ↓ | | | | Random Forgery ↓ | |
|---|---|---|---|---|---|---|---|
| | | 4 vs 1 | 3 vs 1 | 2 vs 1 | 1 vs 1 | 4 vs 1 | 1 vs 1 |
| Sig2Vec Lai et al. (2022) | ChS | 9.03/4.97 | 8.78/4.92 | 9.87/5.16 | 15.10/7.27 | 1.93/0.74 | 5.09/1.18 |
| | TDS | 5.18/2.07 | 5.24/2.22 | 5.94/2.17 | 7.01/3.26 | 1.66/0.26 | 1.76/0.28 |
| | Both | 5.04/1.83 | 5.23/1.83 | 5.28/1.78 | 8.89/2.96 | 0.63/0.12 | 1.42/0.20 |
| DsDTW Jiang et al. (2022) | ChS | 5.91/2.90 | 5.69/2.90 | 5.96/2.77 | 9.58/3.99 | 0.84/0.11 | 1.87/0.17 |
| | TDS | 4.13/1.42 | 4.05/1.41 | 4.40/1.32 | 5.76/1.85 | 0.42/0.07 | **0.59**/0.14 |
| | Both | 3.77/0.89 | 3.65/0.93 | 3.80/1.03 | 6.22/2.08 | **0.15/0.03** | 0.94/0.16 |
| SPECTRUM (**Ours**) | ChS | 5.30/2.47 | 5.33/2.53 | 5.88/2.62 | 10.70/4.97 | 0.72/0.11 | 2.72/0.32 |
| | TDS | 3.38/1.20 | 3.48/1.11 | 3.57/1.18 | 5.20/2.10 | 0.30/0.04 | 0.76/**0.02** |
| | Both | **3.15/0.80** | **3.11/0.81** | **3.23/0.78** | **5.76/1.25** | 0.21/0.05 | 1.08/0.06 |

near-top results in random forgery verification. In this context, SPECTRUM achieves multimodal learning not only across feature domains (temporal and frequency) but also biometric domains, bolstering verification performance through the unprecedented synergy of feature and biometric modalities. (3) In experiments, even simple concatenation or averaging of representations extracted from ChS and TDS could yield improved performance. Designing more sophisticated modality fusion mechanisms to delve deeper into the commonalities between two handwritten biometrics could further enhance model outcomes, pointing out a promising future direction.

## 5 LIMITATION AND DISCUSSION

Although SPECTRUM achieves optimal or SOTA-comparable performances on three datasets, the performance enhancement on Chinese/Latin signatures is less pronounced than on Token Digit String (TDS). This calls for further efforts to improve the generalizability of temporal-frequency multimodal learning on diverse handwritten data types. Additionally, our exploration of multimodal learning has hitherto been confined to temporal and frequency domains. However, it is possible to investigate other modalities such as the spatial modality (rendering online data to offline images) and the video modality (capturing hand movements during writing), as well as the integration of more than two feature modalities, to further enhance the robustness of handwriting verification.

Furthermore, the successful integration of multiple handwritten biometrics points out another simple yet effective avenue to improve OHV performance, with potential benefits for real-world applications such as banking. Despite its straightforwardness, this approach remains unexplored, and available datasets are scarce. This underscores the need for further exploration in this area, such as using a broader range of handwritten biometrics beyond just signature and TDS, collecting more comprehensive multi-biometric datasets, developing specialized techniques for more effective biometric merging, and integrating handwritten biometrics with other behavioral biometrics (*e.g.*, face, fingerprint).

## 6 CONCLUSION

In this paper, we propose SPECTRUM, a novel OHV model driven by multimodal representation learning. We propose a multi-scale interactor for blending local temporal and frequency features across multiple spatial scales, coupled with a self-gated fusion module that integrates global temporal and frequency features through a self-balance. In addition, a multimodal distance-based verifier is proposed, which naturally harnesses both temporal and frequency representations in the multimodal context to sharpen the distinction between genuine and forged samples. Extensive experiments demonstrate the superior performance of SPECTRUM over existing OHV methods, underscoring its effectiveness in multimodal representation learning. Furthermore, we discover that combining multiple handwritten biometrics essentially results in more discriminatory individual representations and facilitates verification. These findings not only confirm the significance of multimodal representation learning in OHV but also highlight promising future directions in enhancing the reliability and applicability of OHV technologies.

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

# APPENDIX

## A  DATA PREPORCESSING

Table 6: Time-function features.

| # | Features |
|---|----------|
| 1-2 | Horizontal and vertical component velocity $x, y$: $\dot{x}, \dot{y}$ |
| 3-4 | Line velocity and acceleration: $v = \sqrt{\dot{x}^2 + \dot{y}^2}, \dot{v}$ |
| 5 | Path-tangent angle: $\theta = \arctan \frac{\dot{y}}{\dot{x}}$ |
| 6-7 | Cosine and sine of angle: $\cos\theta, \sin\theta$ |
| 8-9 | Angular velocity and acceleration: $\dot{\theta}, \ddot{\theta}$ |
| 11 | Centripetal acceleration magnitude: $\triangle v = v \cdot \dot{\theta}$ |
| 12 | Total acceleration magnitude: $a = \sqrt{\dot{v}^2 + \triangle v^2}$ |
| 13-15 | Pressure and its first- and second-order derivatives: $p, \dot{p}, \ddot{p}$ |

We utilize the $x$, $y$ coordinates, and pressure $p$ of the raw online handwritten data for further preprocessing. To mitigate variations in size and location, we perform center normalization on $x$ and $y$, relocating the handwriting center to (0,0) and normalizing coordinates to the range of (-1,1) with preserved aspect ratio. A min-max normalization is also applied to the pressure information. Subsequently, following the official papers, we resample the data in MSDS-ChS and MSDS-TDS into 120Hz and the data in DeepSignDB into 100Hz, using bi-cubic interpolation. We extract 15 time-function features based on the normalized $x$, $y$, and $p$ as model input, as outlined in Table 6. The z-score normalization is applied to the time-function features to standardize them with zero means and unit variance in all experiments.

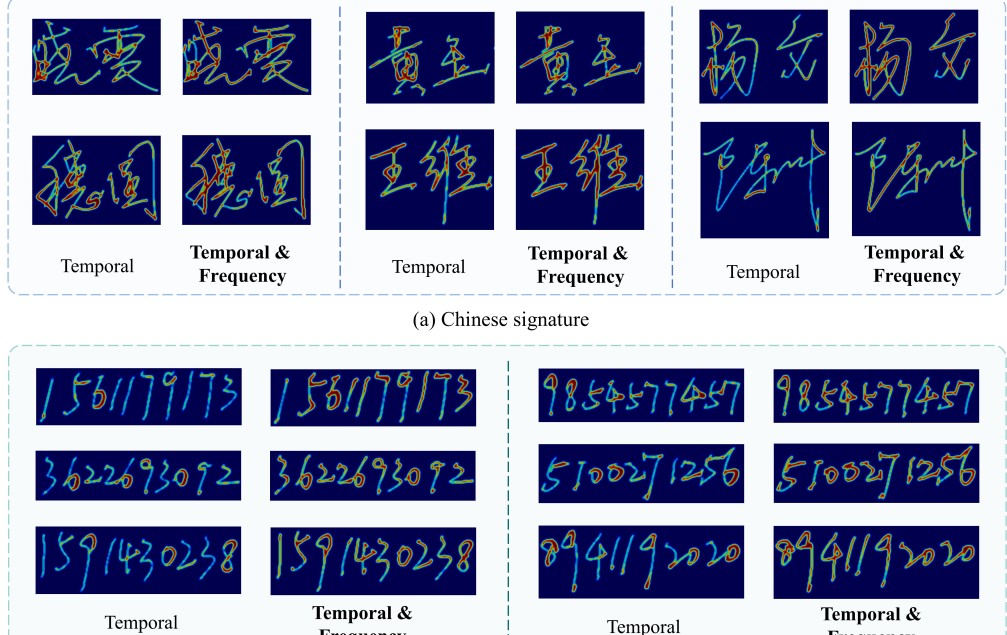

(a) Chinese signature

(b) Token Digit String

Figure 5: Visualization of the final feature representations on Chinese signature and Token Digit String data using MSDS-ChS and MSDS-TDS Zhang et al. (2022). The "Temporal" features are output by the Baseline model (as described in Sec. 4.3) that merely involves temporal domain learning, while the "**Temporal & Frequency**" features are obtained from our SPECTRUM. The handwritten data are desensitized through cropping to protect privacy.

## B  IMPLEMENTATION DETAIL

We train SPECTRUM for 40 epochs, using AdamW Loshchilov & Hutter (2019) with $\beta_1 = 0.9$, $\beta_2 = 0.999$, and weight decay of 1e-2 as the optimizer. The learning rate is initially set to 5e-4 and decreases to 5e-7 following the cosine schedule. In each batch, we randomly sample handwriting from four writers, comprising five genuine samples, five skilled forgeries, and five random forgeries per writer, resulting in a batch size of $4 \times (5 + 5 \times 2) = 60$. Genuine samples and skilled forgeries are drawn from the genuine and skillfully forged data available in the dataset, while random forgeries are randomly selected genuine handwriting of five other writers. $\lambda$ in the loss function is set to 0.1.

## C  VISUALIZATION

To more intuitively demonstrate the effectiveness of the temporal-frequency synergistic learning of SPECTRUM, we visualize the output feature sequence based on single-modal and multimodal learning. Features are extracted from the same handwriting samples for comparison. We utilized the final output features of the Baseline model (as described in Sec. 4.3) for visualization in the temporal domain, while using the output features of the proposed SPECTRUM for visualization in the temporal-frequency domain. Visualizations are presented in Fig. 5, which are performed on the Chinese signature data of MSDS-ChS and Token Digit String data of MSDS-TDS, respectively.

Comparing the left and right columns of each data type, the heatmaps on the right column showcase richer and denser regions with high response values, particularly evident in the Token Digit String data. This suggests that incorporating frequency features with temporal features strengthens the sensitivity of individual writing patterns, resulting in more informative handwriting representations and improved verification accuracy. In addition, as seen in the right-column heatmaps, the high-response regions are concentrated in areas such as stroke twirls, stroke hyphenations, and the start/end of strokes. These regions likely contain richer writing style characteristics, which are effectively captured by the frequency modeling approach. By highlighting these stylistically rich areas, our model demonstrates its ability to focus on crucial elements that distinguish individual writing patterns, further validating the strength of our multimodal approach.

