# OpenReview forum: "SPECTRUM: Empowering Online Handwriting Verification via Temporal-Frequency Multimodal Representation Learning"
_ICLR.cc/2025/Conference — ICLR 2025 Conference Withdrawn Submission_

### Official Review · Reviewer_zgwr · 2024-11-04

**Soundness:** 4
**Presentation:** 3
**Contribution:** 4
**Rating:** 8
**Confidence:** 4

**Summary:**

The paper introduces SPECTRUM, a multi-modal representation model designed to enhance online handwriting verification (OHV) by integrating temporal and frequency features. The model employs a multi-scale interactor for fine-grained temporal and frequency feature interaction, a self-gated fusion module for macro-level integration, and a multimodal distance-based verifier (MDV) that leverages temporal and frequency representations to distinguish genuine handwriting from forgeries. Experimental results across three major datasets (MSDS-ChS, MSDS-TDS, and DeepSignDB) demonstrate SPECTRUM’s superior performance in skilled and random forgery scenarios, setting new benchmarks for OHV accuracy. Furthermore, combining multiple handwritten biometrics, such as Chinese signatures and token digit strings, boosts verification performance, underscoring the model’s versatility and potential for broader biometric applications.

**Strengths:**

- **Innovative Multi-Modal Architecture**: SPECTRUM’s integration of temporal and frequency modalities sets it apart from traditional temporal-only OHV methods, making it highly effective in distinguishing genuine handwriting from forgeries.
- **Effective Verification Across Datasets**: The model’s demonstrated success across different datasets (MSDS-ChS, MSDS-TDS, and DeepSignDB) indicates its robustness and adaptability, with performance gains across Chinese and Latin signature verifications.
- **Advanced Metric for Verification**: The multimodal distance-based verifier (MDV) introduces a more nuanced approach to evaluating handwriting authenticity by leveraging dual modalities, significantly enhancing verification task accuracy.
- **Comprehensive Experimental Analysis**: The authors conduct thorough ablation studies, baseline comparisons, and multimodal tests, offering a holistic evaluation of SPECTRUM’s performance across multiple scenarios.

**Weaknesses:**

- **Scalability and Real-World Feasibility**: Although SPECTRUM demonstrates high performance, the paper could further discuss computational efficiency, especially for large-scale OHV tasks or real-time applications. Time would be beneficial if an analysis of computational cost or inference were included.
- **Limited Exploration of Dataset Biases**: The paper briefly mentions the use of Chinese and Latin signatures but does not explore potential biases that could arise from dataset limitations. A discussion on demographic diversity in handwriting samples would help assess the generalizability of SPECTRUM.
- **Impact of Multi-Scale Integration**: While the multi-scale interactor is effective, the paper could delve deeper into the effects of different scale configurations, particularly for datasets with variable stroke dynamics, such as cursive signatures or complex character-based languages.

**Questions:**

1. Computational Efficiency: Could you provide insights on SPECTRUM's computational efficiency, particularly in real-time scenarios? What optimizations could be applied to enhance its performance?
	2. Diversity in Handwriting Samples: Have you considered or evaluated SPECTRUM’s performance across diverse demographic groups? How might this impact its robustness in real-world applications?
	3. Future Directions for Multi-Modal OHV: Beyond temporal and frequency features, do you envision incorporating additional modalities, such as pressure or spatial data, to enhance verification performance further?

---

### Official Review · Reviewer_v1ku · 2024-11-04

**Soundness:** 3
**Presentation:** 3
**Contribution:** 2
**Rating:** 5
**Confidence:** 4

**Summary:**

This paper proposes SPECTRUM, which is a temporal-frequency synergistic model tailored to enhance handwriting representations to deal with online handwriting verification (OHV) problems. This module comprises three components: a multi-scale interactor that interweaves fine-grained temporal and frequency features; a self-gated fusion module to dynamically integrate global temporal and frequency features; a multimodal distance-based verifier that fully harnesses temporal and frequency representations. The main contributions are as follows: (1) first propose a multimodal representation model for OHV; (2) a multi-scale interactor and self-gated fusion module, designing a multi-modal distance-based verifier; (3) detailed experiments to verify the effectiveness of the proposed SEPCTRUM.

**Strengths:**

1.	The research paper is well-written. The paper is clear and detailed in discussing the main points of the concept with a carefully-designed experiment process which includes results for main datasets and ablation studies. The author also describes the limitations and future research directions in the end of the paper.
2.	The multi-modal learning paradigm for OHV demonstrates a degree of originality. While previous methods typically employed a single modality, focusing on either temporal or frequency aspects, the author proposed a module that effectively fuses these two dynamic modalities.

**Weaknesses:**

1.	Some experiments indicated that the performance of SPECTRUM is suboptimal. For instance, in the evaluation on DeepSignDB (Table 3), SPECTRUM did not outperform previous methods, such as Sig2Vec and DsDTW, in certain aspects. The author did not discuss the reasons for SPECTRUM’s weaker performance on this dataset.
2.	In the experimental process, the methods used for comparison are primarily from 2021 and 2022, which are somewhat outdated. The paper should discuss recent advancements in OHV methods and include comparisons with more recent approaches.
3.	Some parameter choices are unclear, and additional ablation studies are needed. For example, why is $\lambda$ set to 0.1? Are there any ablation studies to support this choice? Similarly, how is the threshold $c$ determined, and are there any justifications or ablation studies provided? Moreover, the loss terms include $L_{intra}$, $L_{tri}$, and $L_{BCE}$; however, the paper does not explain why these particular losses were chosen or if any ablation studies were conducted to validate their selection.

**Questions:**

1.	The author proposed SPECTRUM with a multi-modal learning paradigm; however, the comparison methods used throughout the paper are mostly from before 2022. How has the OHV method evolved in 2023 and 2024? Are there any recent advancements from the past two years?
2.	The paper introduces SPECTRUM, which performs open-set verification, but it is only evaluated in three settings: Chinese signature, token digit string, and Latin signature. Can SPECTRUM generalize effectively to other settings?
3.	Can the SPECTRUM perform better than Transformer-based models?
4.	This paper is well-executed, but some experiments and details are somewhat outdated and lack recent developments. If the author could address these issues thoroughly, I would reconsider the final rating.

---

### Official Review · Reviewer_xJGk · 2024-11-04

**Soundness:** 3
**Presentation:** 3
**Contribution:** 2
**Rating:** 5
**Confidence:** 4

**Summary:**

This submission proposes an online handwriting verification method, SPECTRUM, using temporal- frequency information. Multi-scale interactors are proposed to facilitate finegrained interaction between temporal and frequency features. To balance the weight of temporal and frequency features, a self-gated fusion module is applied. Experiments are conducted on three online datasets -- MSDS-ChS, MSDS-TDS, and DeepSignDB Tolosana, and the results are superior to its peers.

**Strengths:**

+ Combing temporal and frequency information.
+ Performance superior to its peers.
+ Idea is easy to follow.

**Weaknesses:**

- Limited novelty. Fourier transform has been widely used in OHV before. The submission works on some details on using FT.
- Experiments only conducted on the three datasets and lacking comparison on multi-linguistic cases.

**Questions:**

I don't have any question on this kind submission, correct but very limited incremental contribution.

---

### Official Review · Reviewer_vqUX · 2024-11-04

**Soundness:** 2
**Presentation:** 2
**Contribution:** 2
**Rating:** 3
**Confidence:** 5

**Summary:**

This is primarily a paper on online handwriting verification. The main contribution of this paper can be summarized as,

- a multi-scale interactor that integrates fine-grained temporal and frequency features across multiple scales through complementary domain interactions
- a self-gated fusion module, dynamically integrating global temporal and frequency features through self-driven balancing.
- a multimodal distance-based verifier that fully leverages temporal and frequency representations, enhancing genuine-versus-forged discrimination beyond traditional temporal-only methods.

The authors designed several experiments to demonstrate the effectiveness of their approach.

**Strengths:**

The authors have clearly articulated the main focus and innovations of their research, and the experiments have, to some extent, validated the effectiveness of their approach.

**Weaknesses:**

The main concerns, questions, limitations and weaknesses of this paper are as follows:

- In this paper, images and their frequency domain information obtained via DFT are used as input for network training, and this approach is referred to as multimodal features learning by the authors. However, I believe this terminology is inappropriate (although the even and odd time steps are treated separately as temporal and frequency features, respectively). Multimodal typically refers to different types of data sources, such as visual, text, audio, or sensor data. Therefore, the combination of temporal and spatial domains is more accurately described as multi-dimensional or multi-scale features rather than multimodal.

- Could the authors provide a comprehensive analysis of the inference speed during the model testing phase and compare it with previous algorithms?

- More ablation studies should be conducted to validate the effectiveness of the design "$x_{even}$ for temporal and $x_{odd}$ for frequency". Specifically, the experiments that should be supplemented include "$x_{even}$ for frequency and $x_{odd}$ for temporal" and "both $x_{even}$ and $x_{odd}$ for frequency and both $x_{even}$ and $x_{odd}$ for temporal".

- In Tables 1, 2, 3 and 5, the existing algorithms compared by the authors were all proposed at least two years ago. Are there any more recent algorithms available for comparison?

- What is the parameter count for each module in the proposed algorithm? Could the authors provide a comprehensive analysis and compare it with previous algorithms?

- Any ablation studies about the number of multi-scale interactors? Any differences between every interactor (e.g., inputs,  structure) except for the "1D learnable complex weights"? If not, why not directly increase the learnable parameters of single-scale interactor and shared other learnable parameters?

- Any visualization results that can validate the effectiveness of the introduced "1D learnable complex weights"?

- How much performance improvement does the introduction of multi-head self-attention bring to the model?

- Any visualization results that can validate the effectiveness of the introduced "self-gated fusion module"?

- One peculiar point is that the authors claim frequency and temporal domains as two modalities (a claim I find unreasonable), yet they perform no alignment during feature fusion (like LMM). Instead, they simply learn a weight to fuse them, which supposedly yields good results.

- Why not Linear + softmax for "self-gated fusion module"?

- The formula representations for sequence images are inconsistent in Section 3.1 and 3.2.

- How to set c in Eq. (7) for practical application?

- Any ablations studies on the loss weight in Eq. (8)?

Overall, this is not a sufficiently strong work that could be accepted by ICLR, as its main contribution—combining temporal and frequency features for online handwriting verification—is already commonplace in the field of deep learning. Additionally, modules like the self-gated fusion module, designed to support the proposed temporal and frequency integration strategy, lack novelty (these modules add a significant number of extra parameters and computational load to achieve the performance improvement, which is meaningless).  Finally, many expressions in the paper are also inadequately phrased.

**Questions:**

Please refer to the content in the 'weaknesses' box; it will not be reiterated here.

---

### Note · Authors · 2025-01-03

I have read and agree with the venue's withdrawal policy on behalf of myself and my co-authors.